# Oocyte mitophagy is critical for extended reproductive longevity

**Vanessa Cota[1], Salman Sohrabi[1,2], Rachel Kaletsky[1,2], Coleen T. Murphy[1,2]***

**1** Department of Molecular Biology, Princeton University, Princeton, New Jersey, United States of America,
**2** LSI Genomics, Princeton University, Princeton, New Jersey, United States of America

\* ctmurphy@princeton.edu

## Abstract

Women's reproductive cessation is the earliest sign of human aging and is caused by decreasing oocyte quality. Similarly, *C. elegans'* reproduction declines in mid-adulthood and is caused by oocyte quality decline. Aberrant mitochondrial morphology is a hallmark of age-related dysfunction, but the role of mitochondrial morphology and dynamics in reproductive aging is unclear. We examined the requirements for mitochondrial fusion and fission in oocytes of both wild-type worms and the long-lived, long-reproducing insulin-like receptor mutant *daf-2*. We find that normal reproduction requires both fusion and fission, but that *daf-2* mutants utilize a shift towards fission, but not fusion, to extend their reproductive span and oocyte health. *daf-2* mutant oocytes' mitochondria are punctate (fissioned) and this morphology is primed for mitophagy, as loss of the mitophagy regulator PINK-1 shortens *daf-2's* reproductive span. *daf-2* mutants maintain oocyte mitochondria quality with age at least in part through a shift toward punctate mitochondrial morphology and subsequent mitophagy. Supporting this model, Urolithin A, a metabolite that promotes mitophagy, extends reproductive span in wild-type mothers–even in mid-reproduction—by maintaining youthful oocytes with age. Our data suggest that promotion of mitophagy may be an effective strategy to maintain oocyte health with age.

## Author summary

Female reproductive decline begins in a woman's 30's, and has become an increasingly important area of research as more women delay child bearing. Like women, the nematode *C. elegans* undergoes reproductive senescence starting in mid-life, and its reproductive span is determined by oocyte quality decline with age. One of the hallmarks of aging is an increase in mitochondrial dysfunction that can be driven by aberrant mitochondrial fission and fusion, the main regulators of metabolic function and mitochondrial dynamics. In this study we identified a mechanism that promotes reproductive longevity using mitochondrial fission and mitophagy, the trash compactor of dysfunctional mitochondria. This mechanism led us to test Urolithin A, a common metabolite that promotes mitophagy and extends lifespan in *C. elegans*, as a possible therapeutic to delay reproductive

**Data Availability Statement:** All relevant data are within the manuscript and its Supporting Information files.

**Funding:** This project was funded in part by the Global Consortium for Reproductive Longevity and

Equality through the Buck Institute, GCRLE-0220, to CTM. NIGMS Training grant, T32GM007388, Princeton University Molecular Biology Department. The funders had no role in study design, data collection and analysis, decision to publish, or preparation of the manuscript.

**Competing interests:** The authors have declared that no competing interests exist.

decline. We suggest promotion of mitophagy in oocytes as a new target to extend reproductive longevity through Urolithin A supplementation.

## Introduction

Women's reproductive aging is the first major age-related decline that humans experience [1], with increasing fertility issues, miscarriages, and aneuploid pregnancies that begin to arise in women in their mid-30s [2]. These reproductive aging issues have become increasingly important as the average maternal age rises [3,4]. The main contributor to reproductive decline is the age-associated decline of oocyte quality [5,6] rather than loss of oocytes [7]. The mitochondrion is a prime candidate for influencing reproductive aging: the mammalian oocyte contains the largest mtDNA copy number of any other cell [8,9], and mitochondrial dysfunction has been associated with reproductive aging and human oocyte quality decline [10,11].

Mitochondrial dynamics include fission and fusion, which may also play a role in reproductive health. The main drivers of mitochondrial fusion in mammals are Mfn1 and Mfn2 [12], and FZO-1 in yeast and invertebrates [12,13], while fission is controlled by the conserved dynamin-related protein DRP-1 [14,15]. Mouse oocytes with reduced levels of Mfn1 and Mfn2 show signs of mitochondrial and reproductive dysfunction [16,17]. Levels of Drp1 activity are reduced in aged oocytes, and Drp1 KO mice have reduced fertility due to fewer oocytes reaching maturity [18]. These data support a role for fission and fusion in reproduction, but many unanswered questions remain; specifically, how fission and fusion factors affect reproductive decline with age, and whether specific mitochondrial dynamics mechanisms play a role in reproductive longevity.

*C. elegans* is a useful model to study reproductive aging. As for women, *C. elegans'* reproduction declines with age [19]. At the end of reproduction, *C. elegans* germ cells [20] and oocytes [21] deteriorate, fewer oocytes are fertilized, and fewer fertilized embryos hatch [21,22]. Therefore, as in humans, it is the decline of *C. elegans* oocyte quality, not the number of oocytes, that determines reproductive span. Furthermore, mutants that extend reproductive span have been identified, including the insulin/IGF-1 signaling (IIS) receptor mutant *daf-2* [23]. While longevity and reproductive span extension can be uncoupled, such as in TGF-beta mutants [24], insulin/IGF-1 receptor mutants [25] are both long-lived and have a long reproductive span [23] due to the activation of the downstream FOXO transcription factor DAF-16 [24]. *daf-2* mutants maintain germ cell integrity with age [20], maintain oocyte morphology with age, have fewer unfertilized oocytes, and produce more embryos that reach maturity [21,26]—all markers of high-quality oocytes.

Studies of mitochondria in mammalian and *C. elegans* somatic tissues have led to the prevailing model that the elongated morphology is associated with "youthfulness," while punctate ("fragmented") mitochondria are a sign of dysfunction, a hallmark of aging and age-related disease [27]. Mitochondria in *C. elegans* muscle cells [28–32] and neurons [33,34] are elongated in youth and become punctate with age. *daf-2* and mutants of the IIS pathway maintain the elongated mitochondria structure in body wall muscle and neurons longer than does wild type with age [28,30,31,33,34] (Fig 1L and 1M). These findings support a model in which mitochondrial elongation (fusion) is indicative of healthy, functional mitochondria in somatic tissue. However, whether similar mitochondrial dynamics are also required for oocyte health maintenance had not been explored.

Here we investigated the relationship between mitochondrial morphology, oocyte quality, and reproductive longevity. In contrast to the prevailing models of mitochondrial health, we

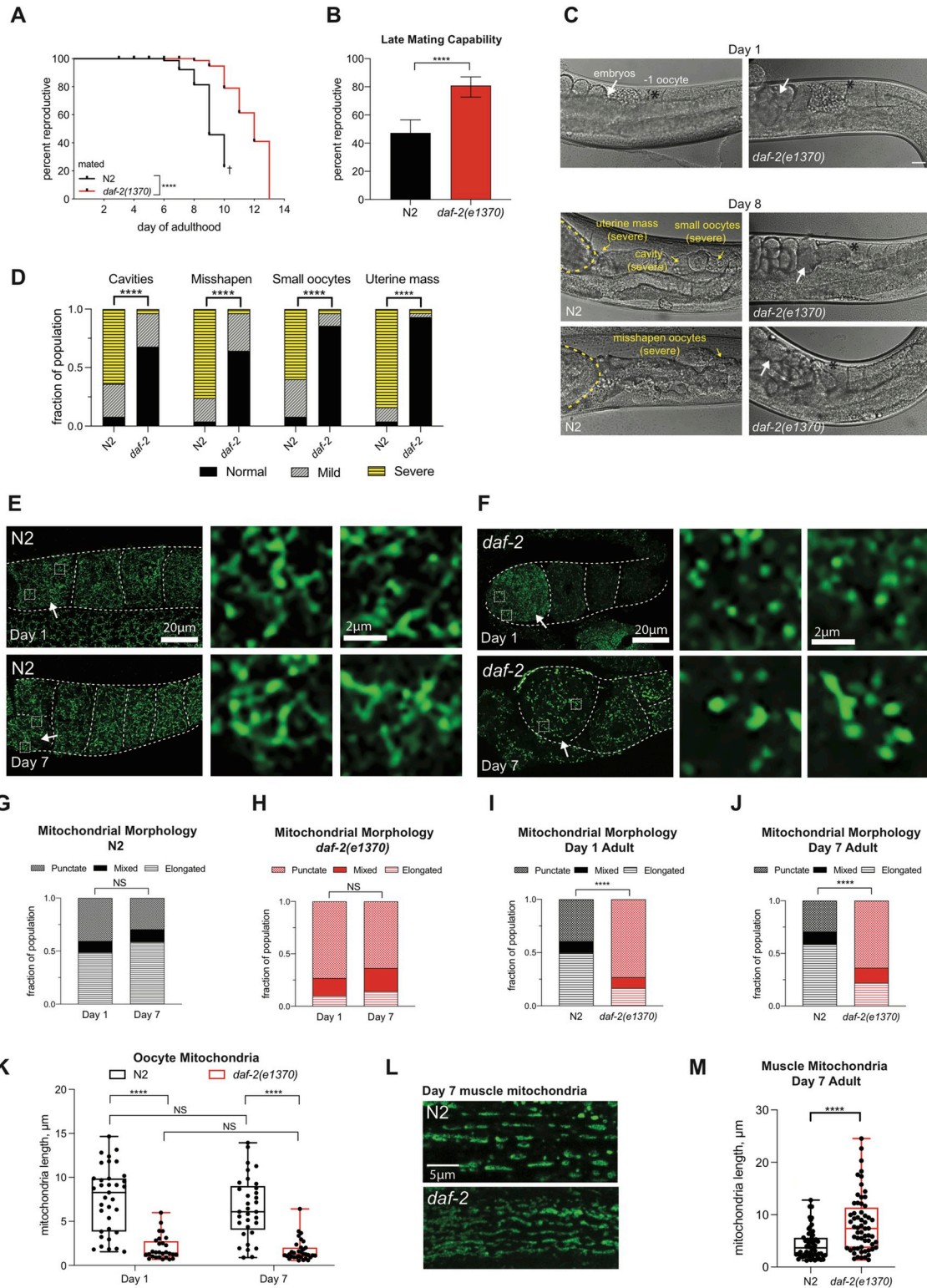

**Fig 1. *daf-2* oocytes display punctate mitochondria.** *daf-2* have an elongated reproductive span and maintain punctate morphology in oocyte mitochondria throughout reproduction. (A) *daf-2(e1370)* reproductive span in hermaphrodites (n = 96–121) mated with males (Luo et al. 2010). † indicates loss of hermaphrodites due to high rates of matricide by bagging. (B) *daf-2(e1370)* (n = 115) and N2 (n = 110) late-mating capability at Day 7 adult. Error bars represent 95% confidence interval. (C) Representative oocyte quality images. N2 (left panel) and *daf-2(e1370)* (right panel) at Day 1 and Day 8 adult. Black asterisks mark the -1 oocyte, white arrows

highlight developing embryos, yellow dotted outline marks uterine mass. Scale bars represent 20μm. (D) Blind scoring of 4 categories of oocyte quality decline in *daf-2(e1370)* and N2 mated Day 8 hermaphrodites (n = 27–30). (E-F) Representative images of mitochondrial morphology using anti-ATP5α to mark mitochondrial membranes. Day 1 and Day 7 adults in N2 (E), and *daf-2 (e1370)* (F). Scale bar 20μm. White-dotted boxes are enlarged sections of the -1 oocyte. Images were processed using the same parameters across the entire image (i.e. adjusting for brightness and contrast). (G-J) Quantification of mitochondrial morphology in the -1 oocyte. Z-stacks were blindly-scored into 3 categories: punctate, mixed, or elongated. Mitochondrial morphology does not change with age in N2 (G, n = 117–83) and *daf-2(e1370)* (H, n = 107–118). *daf-2* have more -1 oocytes with punctate morphology at Day 1(I) and Day 7 (J). Three independent replicates were imaged, scored, and pooled for graphs G-J. Because blind scoring must be compared against the opposing genotype, G and H used the same blind scoring data as I and J. Chi-square test. (K) Mitochondrial length of N2 and *daf-2* was measured using a subset of the same fixed samples used for blind scoring. Each dot represents the average length of 3 individual mitochondrion in one -1 oocyte. One-way Anova. (L) Muscle mitochondria of Day 7 adults using the ATP5a antibody. Images were taken on the confocal. Scale bar 5μm. (M) Muscle mitochondrial length was measured from the same samples used to characterize oocyte mitochondria. Each dot represents 1 of 10 mitochondrion in 6 individual worms for each genotype. Student t-test. ****p $\leq$ 0.0001.

find that the oocytes of the reproductive longevity mutant *daf-2* maintain punctate mitochondrial morphology throughout their extended reproduction–even while their skeletal muscle mitochondria are elongated—while wild-type oocyte mitochondria maintain their elongated morphology with age, even while losing reproductive capacity. Therefore, elongated mitochondrial morphology is uncoupled from youthful function in both *daf-2* and wild-type oocytes. We also find that wild-type oocytes require both fission and fusion for reproductive health, but *daf-2* employs a strategy of fission and subsequent PINK-1-mediated mitophagy to maintain oocyte quality and reproductive longevity. Importantly, we show here that promoting mitophagy through Urolithin A treatment slows reproductive decline in wild-type animals, suggesting that *daf-2's* shift toward fission and mitophagy can be applied to wild-type oocyte mitochondria to improve their oocyte health. Taken together, our data support an alternative strategy for oocyte quality maintenance that engages mitochondrial dynamics and elimination to promote reproductive longevity.

## Results

### *daf-2* oocyte mitochondria are punctate

The insulin-like receptor mutant *daf-2* has an extended lifespan [25], an extended self-fertilized reproductive span [23], and an extended mated reproductive span [24] (Fig 1A). In mated reproductive span assays, hermaphrodites are mated with young wild-type males for 24 hr to avoid convolution with sperm-dependent effects, such as age-impairment of sperm or depletion of self-sperm [22,24] that arise in self-mated (unmated) assays; thus, mated reproductive spans specifically reflect competency of aged oocytes rather than sperm. In a similar test of age-associated reproductive decline, aged *daf-2(e1370)* and wild type (N2) were mated with young males on Day 7 of adulthood ("late mating") to ask whether older animals, at the end of the reproductive span, are reproductively capable and still able to produce progeny. In agreement with the mated reproductive spans, significantly more old (Day 7) *daf-2* worms are able to produce progeny with age (Fig 1B). *daf-2* extends reproductive span by maintaining oocyte quality with age [21,26]. By Day 8 of adulthood, *daf-2(e1370)* mutants still maintain large, uniform, cuboidal oocytes [21], while wild-type oocytes deteriorate [21] (Fig 1C and 1D).

To determine how mitochondria impact oocyte quality and reproductive longevity, we first examined mitochondrial morphology. *daf-2* prevents mitochondrial fragmentation during aging in muscle cells [30,31,35] and neurons [33,34], therefore, we hypothesized that *daf-2* may similarly prevent oocyte mitochondrial fragmentation and maintain elongated mitochondria with age. To test this hypothesis, we examined the mitochondrial morphology of the most mature oocyte, the -1 oocyte, in Day 1 and Day 7 wild-type and *daf-2(e1370)* adults using the

mitochondrial membrane marker ATP5α, which circumvents potential effects on mitochondrial morphology from GFP fusion proteins [36]. Oocytes were imaged and blindly scored for three categories characterizing morphology: elongated (majority of mitochondria in the oocyte are tubular and networked), punctate (majority of mitochondria in the oocyte are small, mainly round, and not interconnected), and mixed (both punctate and elongated forms are represented in the mitochondria population). First, we found that the majority of wild-type oocyte mitochondria appear elongated and tubular in young Day 1 adults, consistent with a previous report [37]. However, this morphology distribution does not change significantly with age or reproductive ability (Figs 1E, 1G, S1A and S1B, S1–S3 Videos). This result suggests that the elongated state of mitochondria in wild-type oocytes does not correlate with "health," as this morphology persists even as the oocytes lose function.

Because high-quality mitochondria are often described as "elongated" while low-quality mitochondria are often described as "fragmented," and *daf-2* skeletal mitochondria are reported to be elongated [30,31] we were also surprised to find that the mitochondria in *daf-2*‐1 oocytes are not primarily elongated and tubular; instead, *daf-2* oocyte mitochondria appear largely punctate in both young (Day 1) and old (Day 7) animals (Figs 1F, 1H and S1A and S1B, S4–S6 Videos), suggesting that rather than fragmenting from an originally elongated state, *daf-2* oocyte mitochondria maintain a punctate state throughout reproduction. Furthermore, *daf-2* mutants have significantly more ‐1 oocytes displaying a punctate morphology than does wild type at either Day 1 or Day 7 (Fig 1I and 1J). To complement the blind scoring results, we measured mitochondrial length; mitochondrial length does not change with age in either N2 or *daf-2* oocytes, but *daf-2*'s oocyte mitochondria are significantly shorter than wild-type's mitochondria on both Day 1 and Day 7 (Fig 1K).

To confirm that skeletal muscle mitochondria morphology was consistent with previously published results [30,31] using the ATP5α antibody, we measured muscle mitochondrial length in the same samples that we used to collect oocyte images. Aged *daf-2(e1370)* muscle mitochondria are significantly longer than aged wild-type muscle mitochondria (Figs 1L and 1M and S1C), as previously reported [30,31]. That is, the mitochondria in oocytes and skeletal muscle of the same animals show distinct morphologies, despite the fact that both muscle and oocytes of *daf-2* animals exhibit extended functional abilities [38]. The punctate morphology maintained with age in *daf-2* oocytes suggests that an alternative mechanism may be employed by oocyte mitochondria to combat age-related reproductive decline, distinct from *daf-2*'s maintenance of elongated mitochondria in aged somatic tissues.

## Fission and fusion maintain reproduction with age

The punctate morphology of *daf-2* oocyte mitochondria is distinct from their morphology in somatic tissues as well as from the morphology of wild-type oocyte mitochondria, which suggests mitochondrial dynamics are differentially regulated in *daf-2* reproduction. Therefore, we investigated the roles of regulators of mitochondrial dynamics in reproduction, DRP-1 (Dynamin-Related Protein 1) and FZO-1 (mitofusin 1/2). *C. elegans* mutants that lack fission (*drp-1 (tm1108)*) or fusion (*fzo-1(tm1133)*) do not exhibit differences in lifespan [32,39] (S2A and S2B Fig), but whether fission and fusion regulate reproductive aging is unknown. The ‐1 oocyte mitochondria of *drp-1* mutants are elongated, while *fzo-1* oocyte mitochondria are punctate (Fig 2A–2C), as expected. The mated reproductive spans of both *drp-1(tm1108)* (Fig 2D) and *fzo-1(tm1133)* (Fig 2E) are significantly shorter than wild type's reproductive span. Knockdown of either *drp-1* or *fzo-1* by RNAi results in a decrease in brood size [40], though whether this progeny loss is due to dysfunctional sperm is unknown. Therefore, we assayed the fission and fusion mutants that were mated with wild-type sperm and found on all the days

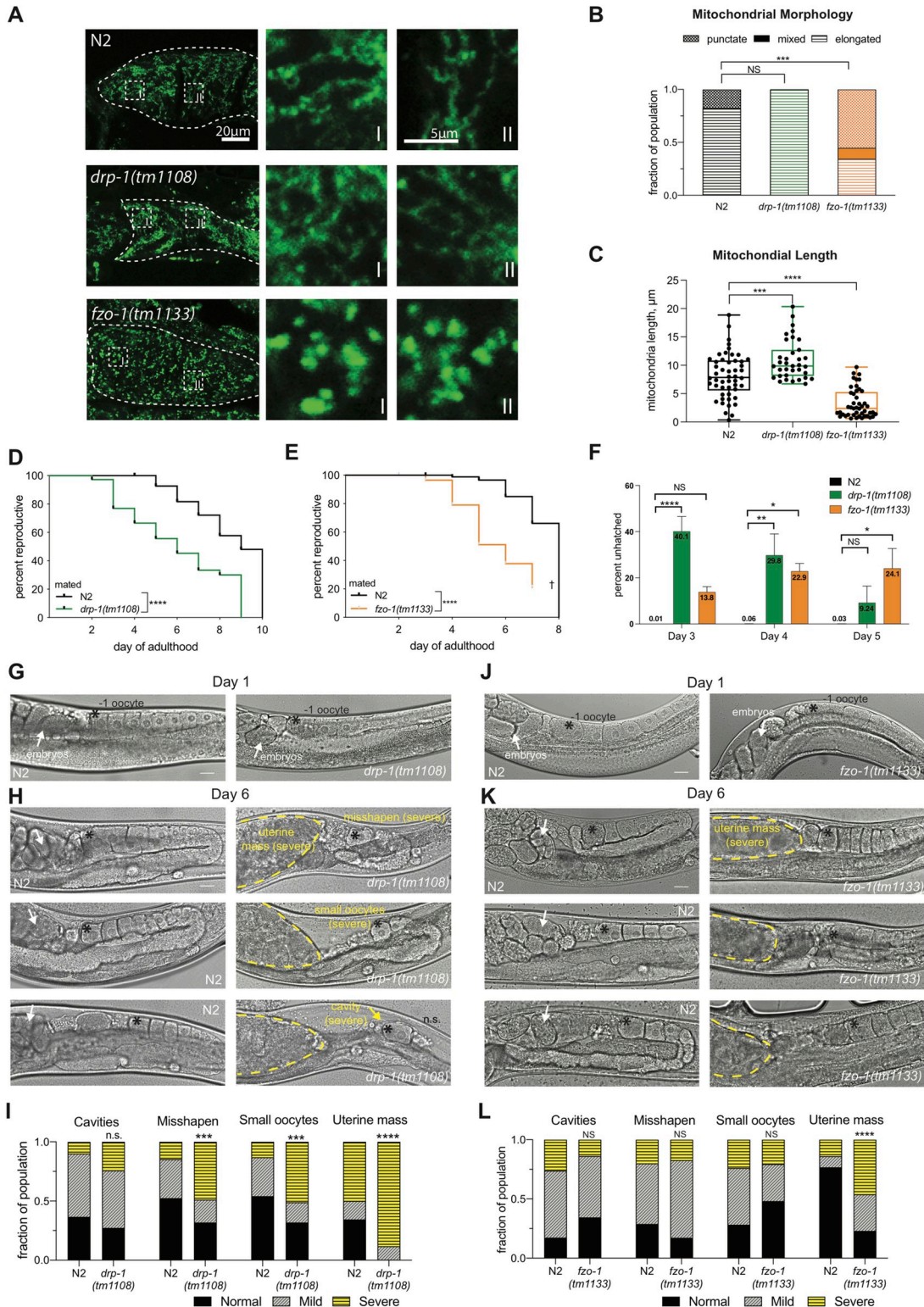

**Fig 2. Fission and fusion are necessary for normal reproductive function with age.** (A) Mitochondrial morphology in mature oocytes of N2, *drp-1(tm1108)*, and *fzo-1(tm1133)* in Day 3 adults. Representative images highlight the different morphologies in N2 (elongated), *drp-1(tm1108)* (elongated), and *fzo-1(tm1133)* (punctate). White-dotted boxes are enlarged (right) of the morphology in the -1 oocyte (I) and the -2 oocyte (II). Images in A were processed using the same parameters across the entire image (i.e. adjusting for brightness and contrast). (B) Blind scoring results of mitochondrial morphology in the -1 oocyte in Day

3 adults of N2 (n = 34), *drp-1(tm1108)* (n = 29), and *fzo-1(tm1133)* (n = 29). (C) Mitochondrial length measurements. Each dot represents 1 of 3 measurements from the -1 oocyte of 20–25 individual hermaphrodites in each genotype. Reproductive spans of mated *drp-1(tm1108)* (D, n = 96–109) and *fzo-1(tm1133)* (E, n = 70–89). (F) Embryonic lethality in N2, *drp-1*, and *fzo-1*. Representative data from days 3, 4, and 5 of adulthood are shown due to high rates of matricide after day 5 (n = 11–13). 2-way Anova. (G-H) Oocyte quality declines with age in *drp-1(tm1108)*. Representative oocyte quality images in Day 1 (G) and Day 6 (H) adults of N2 (left panel) and *drp-1(tm1108)* (right panel). Highlighted features include: embryos in the uterus (white arrows), the -1 oocyte (black asterisk), oocyte quality-decline phenotypes (yellow text and arrows), uterine mass (yellow dotted outline). (I) Quantification of oocyte quality decline phenotypes in N2 (n = 64) and *drp-1(tm1108)* (n = 51). (J-K) Oocyte quality declines in *fzo-1(tm1133)*. Representative oocyte quality images in Day 1 (J) and Day 6 (K) adults of N2 (left panel) and *fzo-1(tm1133)* (right panel). Highlighted features same as H. Scale bars 20µm. (L) Quantification of oocyte quality decline phenotypes in N2 (n = 38) and *fzo-1(tm1133)* (n = 34). Chi-square test. *p ≤ 0.05, **p ≤ 0.01, ***p ≤ 0.001, ****p ≤ 0.0001.

tested the brood sizes of *drp-1(tm1108)* and *fzo-1(tm1133)* are reduced (S2C Fig). In another sign of oocyte quality decline, both the fission and fusion mutants lay significantly more unhatched embryos (Fig 2F). Taken together, these data suggest that disruption in either fission or fusion results in defective oocytes, impairing reproduction.

To determine how loss of mitochondrial fission or fusion affects oocyte quality, we imaged age-matched hermaphrodite germlines and assessed oocyte quality phenotypes. On Day 6 of adulthood, which is near the end of wild type's reproductive span, there is a significantly higher incidence of misshapen and smaller oocytes in *drp-1(tm1108)* fission mutants compared to N2, and many had a mass of oocytes or other cellular material in the uterus ("uterine mass") rather than normally developing embryos (Fig 2H and 2I). Moreover, the oocyte quality phenotypes observed in aged *drp-1(tm1108)* are not present in young adults (Fig 2G) suggesting this decline is dependent on age. Aged *fzo-1* fusion mutant oocytes display severe or mild uterine masses in place of developing embryos in the majority of the population (Fig 2K and 2L). Again, this uterine mass does not appear in young adults (Day 1, Fig 2J). These results suggest that the primary regulators of both fission and fusion are required for normal reproductive health through the maintenance of oocyte quality with age.

## Reproductive longevity requires fission

We next asked whether the punctate state of *daf-2* oocyte mitochondria is necessary for *daf-2*'s extended reproductive span; if *daf-2* specifically requires the punctate morphology in oocytes, then eliminating fission through loss of *drp-1* should prevent *daf-2*'s maintenance of reproduction. We first confirmed that the mitochondria in the -1 oocyte of *daf-2(e1370);drp-1(tm1108)* mutants are elongated, not punctate (Fig 3A–3C). The mated reproductive span and the related late-mating reproductive capability are both significantly reduced in *daf-2(e1370);drp-1(tm1108)* (Fig 3D and 3F), suggesting that the punctate state is required for *daf-2* reproductive span extension. By contrast, knocking out the fusion protein FZO-1, thereby promoting fission resulting in primarily punctate mitochondria like *daf-2* (Fig 3A–3C) does not affect reproduction: loss of FZO-1 does not change *daf-2*'s extended reproductive span (Fig 3E) and does not affect *daf-2*'s reproductive capability in aged hermaphrodites (Fig 3F). Therefore, *daf-2* requires mitochondrial fission, but not fusion, to preserve reproductive function with age.

We next asked whether the requirement for DRP-1 in *daf-2*'s extended reproductive longevity is through the maintenance of oocyte quality. While the "uterine mass" phenotype is not present in younger adults (Day 5, S3A Fig), by Day 8 the majority of mated *daf-2(e1370);drp-1(tm1108)* mutants have developed large masses (instead of developing embryos) in their uteri (Fig 3G and 3H), confirming it as a reproductive aging phenotype. Loss of fusion (*daf-2;fzo-1*) has no effect on *daf-2* oocyte quality (Figs 3I and S3B). Consistent with a decrease in oocyte quality, *daf-2(e1370);drp-1(tm1108)* mutants lay a high percentage of unhatched embryos on days 3, 4, and 5 of adulthood (Fig 3J). Together, our data suggest that retaining the punctate

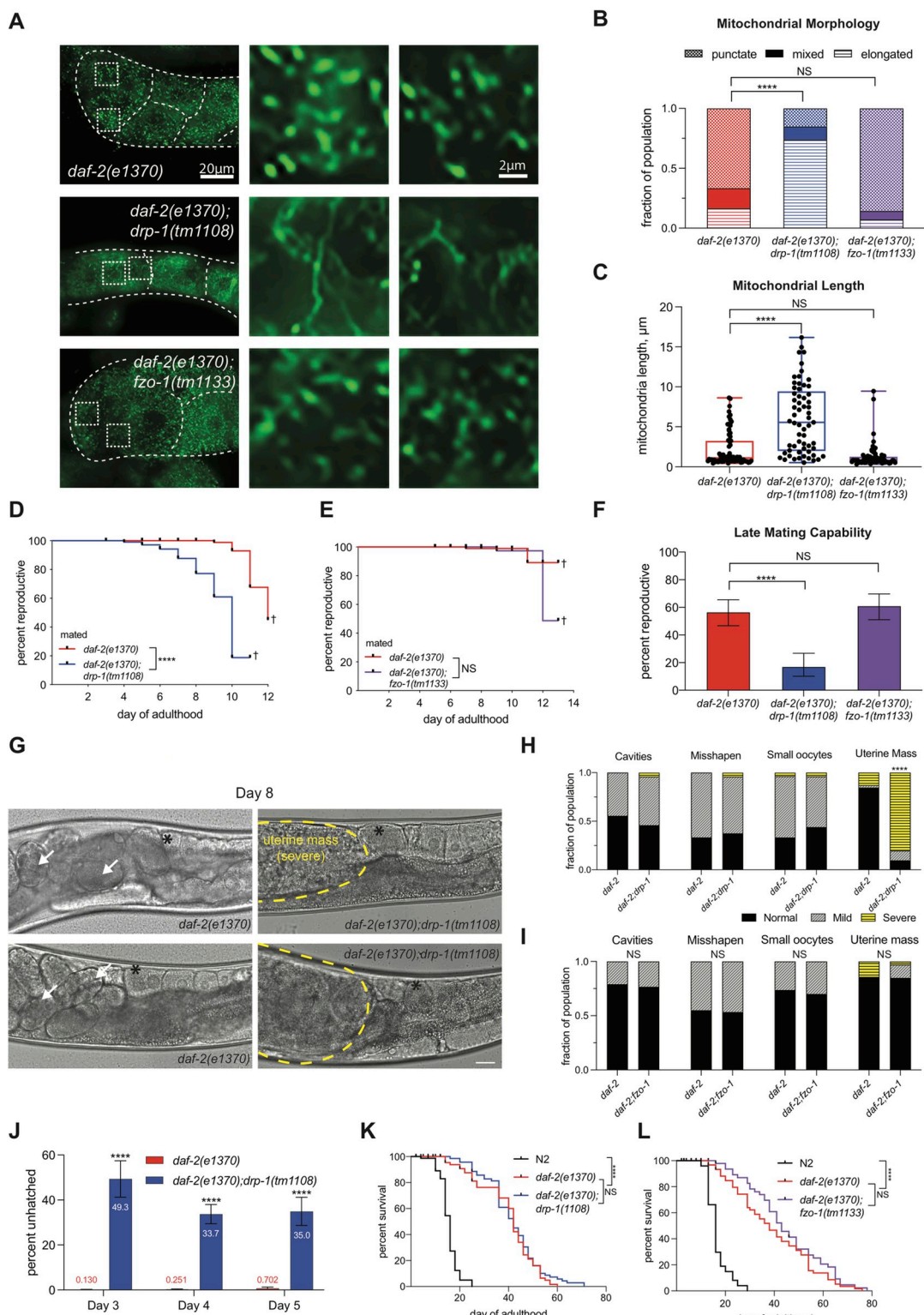

**Fig 3. *daf-2* requires fission, not fusion, for reproductive longevity.** (A) Representative images of Day 4 adult mitochondrial morphology in the -1 oocyte marked with anti-ATP5α. White-dotted boxes are enlarged (right) of the morphology in the -1 oocyte (I and II). Scale bar 10μm. Images in A were processed using the same parameters across the entire image (i.e. adjusting for brightness and contrast). (B) Blind scoring results of mitochondrial morphology of the -1 oocyte in three categories: punctate, mixed, or elongated of *daf-2(e1370)* (n = 30), *daf-2(e1370);drp-1(tm1108)* (n = 46), and *daf-2(e1370);fzo-1(tm1133)*

(n = 56). (C) Mitochondrial length measurements. Each dot is 1 of 3 measurements from the -1 oocyte from 20–25 individual hermaphrodites in each genotype. (D) Fission knockout *daf-2(e1370);drp-1(tm1108)* (n = 96) has a reduced reproductive span compared to *daf-2(e1370)* (n = 121) in mated hermaphrodites. (E) Fusion knockout *daf-2(e1370);fzo-1(tm1133)* (n = 101) maintain an elongated reproductive span compared to *daf-2(e1370)* (n = 102). In D and E † indicates loss of hermaphrodites due to high rates of matricide. (F) Late-mating of *daf-2(e1370)* (n = 103), *daf-2(e1370);drp-1(tm1108)* (n = 79), and *daf-2 (e1370);fzo-1(tm1133)* (n = 102). Error bars represent 95% confidence interval. (G) Representative images of oocyte quality in *daf-2(e1370)* (left panel) and *daf-2;drp*-1 (right panel) in Day 8 adults. Highlighted features include: embryos in the uterus (white arrows), the -1 oocyte (black asterisk), uterine mass (yellow dotted outline). Scale bar is 20μm. (H) Quantification of oocyte quality phenotypes from *daf-2(e1370)* (n = 41) and *daf-2(e1370);drp-1(tm1108)* (n = 39) in Day 8 adults. (I) Oocyte quality does not decline with age in *daf-2;fzo*-1. Quantification of oocyte quality features in *daf-2* (19–21) and *daf-2;fzo-1* (30–33). (J) Embryonic lethality in *daf-2(e1370);drp-1(tm1108)* and *daf-2(e1370)*. Representative data from days 3, 4, and 5 of adulthood are shown due to high rates of matricide after day 5 (n = 11–13). (K) Lifespans of *daf-2(e1370);drp-1(tm1108)* (n = 81) and *daf-2(e1370)* (n = 80) or (L) *daf-2(e1370);fzo-1(tm1133)* (n = 80) and *daf-2(e1370)* (n = 81) are unchanged. $^{**}p \leq 0.01$, $^{****}p \leq 0.0001$.

state of mitochondria through fission is required for *daf-2's* maintenance of high-quality oocytes with age, which in turn is critical for its extended reproductive span.

Despite the dramatic effect on *daf-2* oocyte quality and reproductive span, neither loss of DRP-1 or FZO-1 affected the extended lifespan of *daf-2* mutants (Fig 3K and 3L), suggesting that mitochondrial dynamics specifically affect reproductive longevity through oocyte quality maintenance with age.

### *daf-2's* extended reproductive longevity requires mitophagy

Mitochondria that have accumulated damage divide asymmetrically through fission [41] and activate mitophagy to eliminate mitochondria that are beyond repair [41]. One possible benefit to *daf*-2 maintaining punctate mitochondria in oocytes is they may be primed for mitophagy, which provides quality control for damaged mitochondria. For example, study of induced mitochondrial DNA damage in Drosophila early oogenesis revealed that mitochondria undergo fragmentation to selectively remove mitochondria with mutated mtDNA through mitophagy [42], and mid-life induction of Drp1-mediated skeletal muscle mitochondrial fission and lifespan extension in Drosophila require autophagy/mitophagy [43]. We used the SKI LODGE system [44] to generate a mitophagy reporter (mtRosella) [29,45,46], confirmed it was correctly inserted behind the germline-specific *pie-1* promoter, but found that it is silenced in the germline (S4A Fig). Therefore, we instead used a genetic approach to ask whether mitophagy is required for *daf-2's* extended reproductive span. The main mitophagy pathway activator in *C. elegans* is PINK-1 (PTEN induced kinase 1), which is conserved in yeast, flies, worms, and humans, and functions by accumulating on dysfunctional mitochondria [47–49]. We found that aged *daf-2(e1370);pink-1(tm1779)* double mutants have reduced reproductive capability after late (Day 7) mating (Fig 4A). (*daf-2;pink-1* mutants exhibit high rates of matricide, preventing the analysis of full reproductive spans; however, both mutants have similar mating ability (S4B Fig), suggesting the reduction in reproductive ability is an aging phenotype.) Further, similar to *daf-2(e1370);drp-1(tm1108)* mutants, mated Day 8 *daf-2;pink-1* hermaphrodites exhibited a significant number of uterine masses rather than developing embryos (Fig 4B and 4C; compare to 3G and 3H), suggesting that the uterine mass phenotype is a result of mitochondrial dysfunction in aging oocytes brought on by the loss of mitophagy. There is no difference between the reproductive spans of N2 and *pink-1(tm1133)* (Fig 4D) or late-mating capability (S4C Fig), suggesting that PINK-1 may be specifically required for reproductive longevity mechanisms that favor the elimination of damaged, fissioned mitochondria. Further, like fission and fusion regulators, PINK-1 is not required for somatic longevity in *daf-2* (Fig 4E) or wild type (S4D Fig). Taken together, our results support a model in which punctate

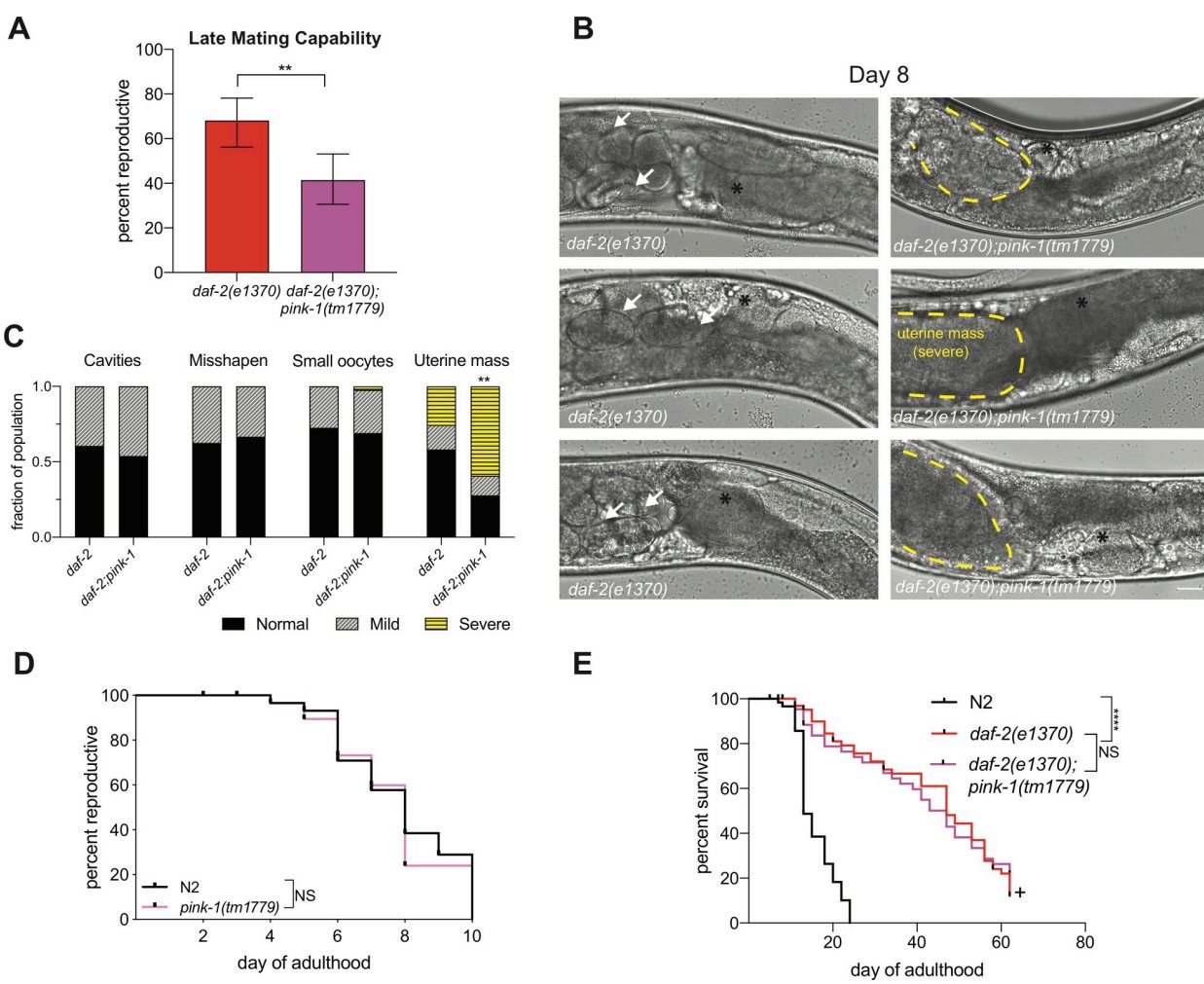

**Fig 4. Mitophagy is required for *daf-2*'s extended reproductive capability.** (A) Late-mating of *daf-2(e1370)* (n = 70) and *daf-2(e1370);pink-1 (tm1779)* (n = 66). Late-mating performed on age-matched Day 7 hermaphrodites provided with young males. Error bars represent 95% confidence interval. (B) Representative images of oocyte quality assay of *daf-2(e1370);pink-1(tm1779)* in aged adult germlines (Day 8). Highlighted features include: embryos in the uterus (white arrows), the -1 oocyte (black asterisk), uterine mass (yellow dotted outline). Scale bar is 20μm. (C) Quantification of oocyte quality phenotypes in *daf-2(e1370)* (n = 45) and *daf-2(e1370);drp-1(tm1108)* (n = 51). (D) There is no significant difference between the mated reproductive spans of N2 (n = 59) and *pink-1(tm1779)* (n = 63). (E) No change between the lifespan of *daf-2(e1370); pink-1(tm1779)* compared to *daf-2(e1370)* alone (n = 66–79). + indicates censorship due to covid-19 shutdown. **p ≤ 0.01, ****p ≤ 0.0001.

mitochondria are primed for mitophagy specifically in *daf-2* oocytes to promote reproductive longevity.

## Urolithin A slows reproductive decline

Given that *daf-2's* reproductive span is double that of wild type's and that *daf-2* requires mitophagy, we asked whether wild-type animals' reproduction could be improved by promoting mitophagy. Urolithin A (UA) is an ellagitannin metabolite [50,51] that increases mitophagy through the accumulation and stabilization of PINK-1 [46,52]. UA increases lifespan in worms [52] through the upregulation of mitophagy [46,52,53], improves muscle function in aged mice [52], and increases mitophagy in muscular dystrophy mouse models and human patients [53,54]. Besides muscle function, Urolithin A has been found to improve the function of a variety of tissues and organs through mitophagy upregulation [55]; however, whether UA supplementation

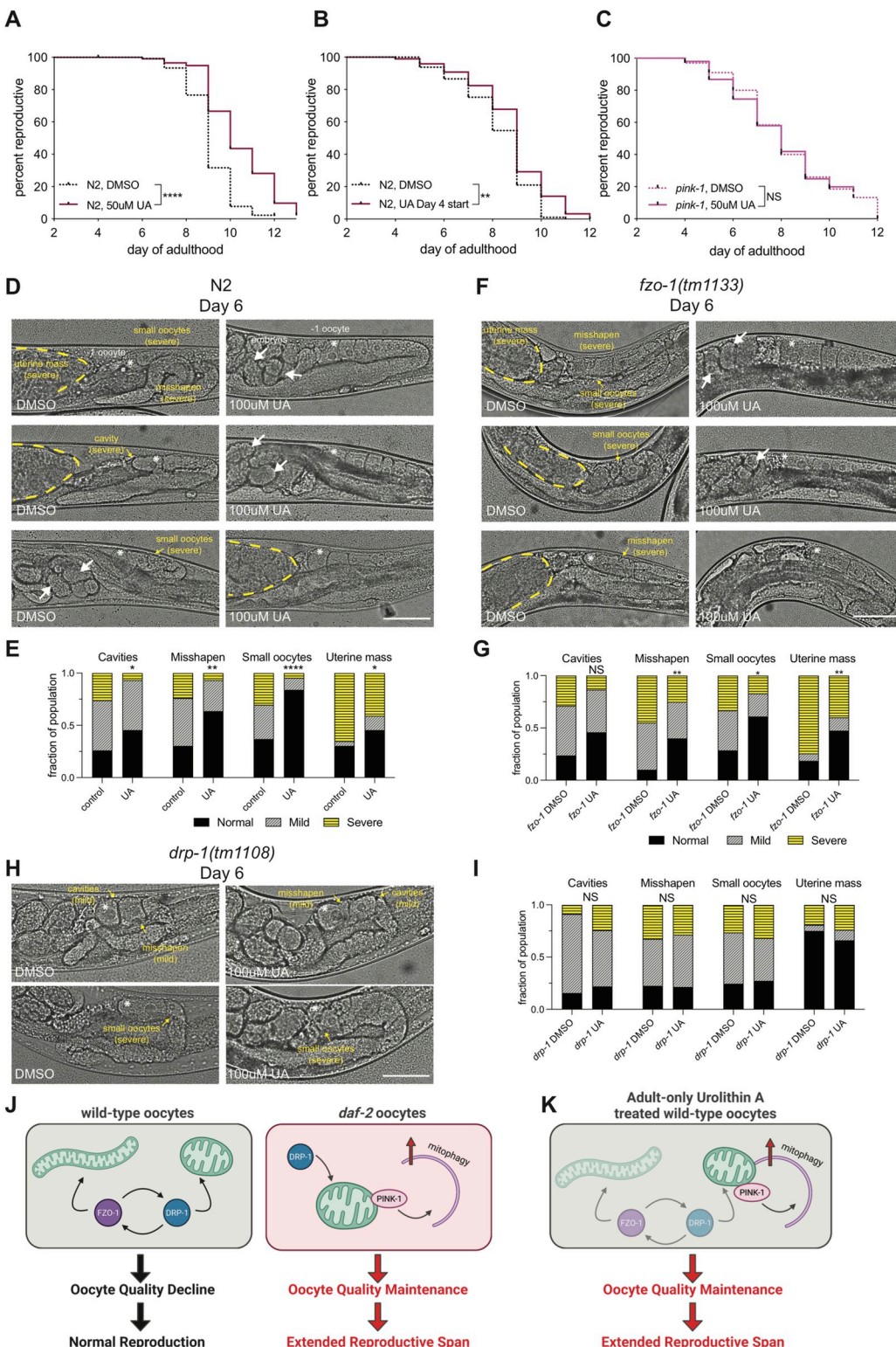

**Fig 5. Urolithin A treatment slows reproductive decline in wild type.** (A) Reproductive span of N2 control 0.3% DMSO (n = 100) and N2 50μM UA, 0.3% DMSO (n = 100). Treatment started on Day 1 of adulthood. (B) Reproductive span of N2 hermaphrodites with Urolithin A treatment starting at Day 4 of adulthood is improved. Both control (n = 100) and experimental (n = 100) groups began 0.3% DMSO at Day 1 and UA treatment began at Day 4 in the experimental group. p = 0.0062 (C) Reproductive span of *pink-1(tm1779)* is unchanged with UA treatment. Control 0.3% DMSO (n = 100) and

experimental 50μM UA, 0.3% DMSO (n = 100). In A and C mated hermaphrodites were loaded into Celab and started treatment on Day 1. (D) Representative images of oocyte quality in control (left panel) and UA treated (right panel) N2 worms at Day 6. Highlighted features include: embryos in the uterus (white arrows), the -1 oocyte (white asterisk), and oocyte quality decline phenotypes (yellow text). (E) Quantification of oocyte quality phenotypes at Day 6 of adulthood in control (n = 53–55) and UA treated groups (n = 48–50). (F) Representative images of oocyte quality in control (left panel) and UA treated (right panel) *fzo-1(tm133)* worms at Day 6. Highlighted features include: embryos in the uterus (white arrows), the -1 oocyte (black asterisk), and oocyte quality decline phenotypes (yellow text). (G) Quantification of oocyte quality phenotypes at Day 6 of adulthood in *fzo-1* control (n = 38–43) and *fzo*-1 UA treated groups (n = 37–41). (H) Representative images of oocyte quality in control (left panel) and UA treated (right panel) *drp-1(tm1108)* worms at Day 6. (I) Quantification of oocyte quality phenotypes at Day 6 of adulthood in *drp-1* control (n = 52) and *drp-1* UA treated groups (n = 50). (J) Wild type requires both fission and fusion, but not mitophagy, to maintain a normal reproductive span. Reproductive longevity in *daf-2* requires fissioned mitochondria, which are then primed for and utilize mitophagy to maintain reproductive longevity through oocyte quality maintenance. (K) Urolithin A, a mitophagy promoter, extends reproductive span in wild type with adult only treatment at Day 1 or Day 4 of adulthood. *$p \leq 0.05$, **$p \leq 0.01$, ***$p \leq 0.001$, ****$p \leq 0.0001$.

affects age-related reproductive decline is unknown. Therefore, we asked whether Urolithin A (UA) supplementation would increase the reproductive span in wild-type *C. elegans*.

Supplementing mated wild-type hermaphrodites with UA at the start of reproduction (Day 1) significantly extended reproductive span (Fig 5A; 14% increase, p<0.0001). Furthermore, when worms are supplemented with UA at the start of reproductive decline (Day 4), late-start UA treatment still significantly extended the reproductive span of wild-type animals (Fig 5B), suggesting UA treatment acts quickly to improve oocytes even when given in middle-aged adults. Treatment of mated *pink-1(tm1779)* mutants with UA did not increase reproductive span (Fig 5C), suggesting that UA-mediated reproductive span extension requires PINK-1-mediated mitophagy.

Day 6 adult-only UA-treated worms had significantly more large and uniformly shaped oocytes, which are signs of youthfulness (Fig 5D and 5E). Further, in *fzo-1* mutants, in which the majority of mitochondria are punctate due to loss of fusion (Fig 2A–2C), oocyte quality is improved with UA treatment and the uterine mass phenotype seen in *fzo-1* mutants alone (Fig 2J and 2K) is partially corrected (Fig 5F and 5G). By contrast, Urolithin A treatment of *drp-1* did not significantly improve oocyte quality decline phenotypes (Fig 5H and 5I). Therefore, while we cannot solely attribute the positive effects of UA treatment to its function in oocytes, as drug treatment is administered to the whole animal and therefore might affect non-oocyte tissues, we can conclude that adult treatment with Urolithin A can improve oocyte quality maintenance with age and subsequently extend the reproductive span of wild-type animals. Urolithin A might function by shifting the mitochondrial maintenance strategy to mitophagy, as *daf-2* mutants do through PINK-1 function.

Here we have uncovered a mitochondrial mitophagy-based mechanism to promote reproductive longevity. We found that *daf-2*'s reproductive span extension and oocyte quality maintenance require mitochondrial fission (DRP-1) and mitophagy (PINK-1) (Fig 5J). Further, our data suggest that mitophagy is a main player in oocyte quality maintenance and reproductive span extension. In support of this conclusion, Urolithin A, a metabolite that promotes mitophagy through PINK-1, improves the reproductive span in wild-type worms by maintaining oocyte quality (Fig 5K).

## Discussion

We sought to determine whether mitochondrial morphology influences oocyte quality and reproductive span, and whether there is an ideal morphology that promotes reproductive health. *daf-2* appears to use a strategy of promoting fission and subsequent mitophagy to increase reproductive longevity, requiring DRP-1 to tip the balance of mitochondria to the

fissioned, punctate state. This result was surprising, as punctate mitochondria are usually thought to be dysfunctional, and are often referred to as "fragmented." However, in the case of *daf-2's* reproduction, punctate mitochondria are beneficial. This benefit may be due to the fact that fissioned mitochondria activate the mitophagy pathway [56]; indeed, we find that mitophagy is required for *daf-2*'s extended reproductive span. In this model, dysfunctional mitochondria may be removed from the mitochondria pool through PINK-1-mediated mitophagy, and this mechanism assists in slowing oocyte quality decline, thereby extending reproduction (Fig 5J). Moreover, applying this strategy to wild-type oocyte mitochondria through the use of Urolithin A improves oocyte quality with age and extends reproductive span.

Insulin/IGF-1 receptor mutants retain elongated mitochondria in muscle and neurons later in life [28,30,33,34], which supports the model that fusion promotes longevity. However, promoting fusion, either by knocking out the fission protein DRP-1 or by overexpressing the fusion protein FZO-1, has no effect on lifespan [32] (S2A and S2B Fig). These data suggest that neither fusion nor fission is the main driver of somatic lifespan. Instead, we have shown here that mitochondrial morphology and dynamics specifically regulate *C. elegans* reproductive span. In fact, most mitochondria (95%) [57] in the *C. elegans* adult is present in the germline, not somatic tissues, and oocyte production results in up to a 6-fold increase in the mtDNA copy number [58]. The tissue-specific effect that mitochondrial dynamics has on reproduction is mirrored in *daf-2* mutants: the loss of *drp-1* and *fzo-1* from *daf-2(e1370)* has no effect on lifespan (Fig 3K and 3L). PINK-1-mediated mitophagy also appears to play no role in *daf-2* somatic longevity (Fig 4E), although autophagic activity [29,59], mitophagy, and mitochondrial biogenesis mechanisms are upregulated in *daf-2* mutants [29]. These results suggest that somatic lifespan and reproductive span have distinct requirements for mitochondrial morphology and employ different, tissue-specific mechanisms to combat age-related mitochondrial dysfunction. These distinct mechanisms may also be regulated differently, as our expression analysis [26] suggests these proteins are not regulated transcriptionally in oocytes. Together, our data suggest that the requirement of mitochondrial dynamics specifically for reproduction, rather than somatic longevity, reflects the need for high-quality oocytes for successful progeny development. Contrary to the view that tubular, elongated mitochondria are healthier than fissioned, punctate mitochondria, the pro-fission and PINK-1-mediated mitophagy mechanism used by *daf-2* mutants to maintain oocyte quality with age is an alternative strategy to promote reproductive longevity.

By using *daf-2* as a model, we have found that promoting mitophagy in wild-type oocytes can help improve oocyte quality and slow reproductive decline (Fig 5A and 5B, 5D and 5E). Extending wild type's reproductive span through oocyte quality maintenance with Urolithin A supplementation supports the model that enhanced mitophagy at least in part promotes reproductive span extension. Our data show that not all mitochondria in wild-type oocytes are elongated, and some punctate mitochondria are present in mature oocytes. Therefore, promoting the stabilization of PINK-1 would allow the activation of mitophagy on punctate/fissioned mitochondria in wild-type oocytes. In further support of this model, when oocyte mitochondria are primarily in the punctate state, as in the fission mutant *fzo-1*, we see an improvement of oocyte morphology with age (Fig 5F and 5G).

Mechanisms to slow reproductive decline have become increasingly important as the average maternal age increases [1–6]. The role of mitophagy in *daf-2's* extended reproductive span provides an alternative strategy to combat age-related reproductive decline and supports investigating mitophagy as a means to promote reproductive health with age. Urolithin A (UA) was recently identified to improve lifespan and healthspan by combating mitochondrial dysfunction and age-related decline via upregulation of mitophagy [52]. Further, Urolithin A has been found to be safe and effective in humans and improves muscle strength [53,60]. A recent '*in*

*vitro* aging' study of bovine oocytes suggests that UA may be beneficial for mammalian *in vitro* ART applications [61]. Our *in vivo* studies of reproductive aging suggest that UA may improve reproductive span with age by promoting maintenance of oocyte quality, supporting our model that mitophagy may be a key component to combating reproductive decline. Further, we show improvement to reproduction can begin with UA treatment in both early and mid-reproductive adulthood (Day 4 in worms, which is equivalent to approximately 35 years in women). UA supplementation has been found to have a positive effect on a variety of tissues and age-related decline phenotypes [55]. Our data suggest that UA may also have potential therapeutic benefits to reproductive health in other organisms.

## Experimental model and subject details

### *C. elegans* strains

The following strains were used in this study: N2 Bristol strain as wild type worms, CB4108: *fog-2(q71) V* males for mating experiments, CB1370: *daf-2(e1370)*, CU6372: *drp-1(tm1108)*, CU5991: *fzo-1(tm1133)*, *pink-1(tm1779)*. The following crosses were performed: CQ613: *daf-2(e1370);drp-1(tm1108)*, CQ611: *daf-2(e1370);fzo-1(tm1133)*, and CQ659: *daf-2(e1370); pink-1(tm1779)*.

## Method details

### General worm maintenance

All strains were cultured using standard methods [62]. For all experiments, worms were maintained at 20˚C. Nematode growth medium (NGM: 3 g/L NaCl, 2.5 g/L Bacto- peptone, 17 g/L Bacto-agar in distilled water, with 1mL/L cholesterol (5 mg/mL in ethanol), 1 mL/L 1M CaCl$_2$, 1 mL/L 1M MgSO$_4$, and 25 mL/L 1M potassium phosphate buffer (pH 6.0) was added to molten agar after autoclaving. Plates were seeded with OP50 *E. coli* for *ad libitum* feeding. To synchronize experimental groups, gravid hermaphrodites were used to collect eggs by treating them with a 15% hypochlorite solution (e.g., 8.0 mL water, 0.5 mL 5N KOH, 1.5 mL sodium hypochlorite) and vortexing, followed by repeated washing of collected eggs in M9 buffer [62].

### Germline mtRosella strain

The TOMM-20::mtRosella insert was isolated from the P$_{myo-3}$TOMM-20::mtRosella (IR1631) strain [29]. We then used the SKI LODGE [44] co-CRISPR strategy, which uses a single-copy tissue specific promoter for transgenic cassette insertions. Using the germline-specific *pie-1* promoter strain (WMB1119), we injected the co-CRISPR mix and confirmed the P$_{pie-1}$TOMM-20::mtRosella via sequencing. Next, we dissected and imaged germlines using the Nikon eclipse Ti at 60x magnification and took Z-stack images in 0.7 μm steps. The images shown in the supplement are superimposed Z-stacks of each fluorescent channel.

### Reproductive span and lifespan assays

Mated reproductive spans were performed as previously described [24]. Eggs were synchronized with a hypochlorite solution and selected for experiments at the L4 stage. Hermaphrodites were mated with *fog-2(q71)* males at a 2:1 ratio for 24 hrs from L4 to Day 1 of adulthood, then were removed from males to prevent further mating. Each hermaphrodite was singled onto a 35mm NGM plate and moved to new plates every 24 hours until the end of the reproductive span. 48 hours after removal, plates were screened for progeny to confirm reproductive span, including male progeny to confirm a successful mating. Reproductive cessation was defined as the last day of progeny production preceding two full days without progeny. If

bagging was observed, hermaphrodites were censored on the day of matricide (defined as progeny hatched within mother). All reproductive spans were performed at 20˚C.

Lifespan assays were performed as previously described [25]. In brief, groups of hypochlorite-synchronized hermaphrodites (<15 per plate) were placed on plates at the L4 stage. The hermaphrodites were transferred to freshly seeded plates every two days when producing progeny, and 2–3 days thereafter. Worms were censored on the day of matricide (defined as progeny hatching within mother), abnormal vulva structures, or loss. Worms were defined as dead when they no longer responded to touch. All lifespans were performed at 20˚C.

### Oocyte quality assays

Assays were performed as previously described [21,26]. Hypochlorite-synchronized hermaphrodites were mated for 24 hrs on either L4 or Day 1 of adulthood. Mating was confirmed as in the reproductive span by confirming the presence of male progeny. Groups of hermaphrodites were maintained until the day of the experiment (Day 6-Day 8 of adulthood); hermaphrodites were added to slides with 3–4% agarose pads in M9 with levamisole. DIC images were captured on a Nikon eclipse Ti at 60x magnification. The scorer was blinded to genotype and each image was given a score (normal, mild, severe) in each oocyte quality category. Note that in experiments using *daf-2(e1370)* background, worms were added to plates with serotonin for 1 hour on Day 6 and Day 8 to purge embryos and prevent matricide.

### Late-mating assay

Late-mating assays were performed as previously described [63]. Hypochlorite-synchronized worms were selected at the L4 stage. Hermaphrodites were aged to Day 7, moving every two days to separate from progeny. On the day of late mating, hermaphrodites were individually placed onto 35mm plates seeded with 25uL spots of OP50. *fog-2(q71)* males were added at a 3:1 ratio and the presence of progeny was scored 4–5 days after males were added to the plates.

### Embryonic lethality assay and progeny counts

Assay was performed as described [21]. L4 hermaphrodites were singled onto 35mm NGM plates and males were added at a 3:1 ratio for mating. Mating was confirmed as in the reproductive span, and mated hermaphrodites were removed from males by transfer to a new plate. Every 24 hours, hermaphrodites were moved to new plates. Approximately 24 hours after removal, plates were scored for numbers of progeny, unhatched embryos, and unfertilized oocytes. Counting continued throughout reproductive span until the hermaphrodites' numbers fell below 10. Percent unhatched is the proportion of embryos that failed to hatch after 24 hours.

### Mitochondria morphology

Mitochondria morphology was assessed using immunohistochemistry staining with the mitochondria membrane marker ATP5α. Germline immunostaining was performed using the protocol as in [64]. In brief, hypochlorite-synchronized hermaphrodites were maintained until the day of interest. Hermaphrodite germlines were dissected and fixed using a methanol/acetone fixation. Primary antibody was from AbCam Mouse Anti- ATP5α antibody [15H4C4], and secondary was Invitrogen Goat Polyclonal Secondary Antibody Alexa Fluor Plus 555. Imaging was performed on point scanning confocal Nikon A1 at either 100x or 60x or Nikon eclipse Ti. Images were taken on a Z-stack (0.7–1.0 μm steps) focusing on the -1 oocyte. For mitochondria morphology quantification, images from 2–4 genotypes were collected and file

names were blinded. Each Z-stack was then assessed for a morphology qualification (punctate, mixed, elongated). Images were processed in FIJI and Adobe Photoshop. Each figure of images shown were from the same experiment, imaged on the same microscope at the same magnification, and with the same exposure. All images in each figure were processed in the same way across the entire image (e.g., adjusting for brightness and contrast).

## Urolithin A treatment reproductive and lifespans

Urolithin A was obtained through Fisher Scientific and diluted in DMSO. The control is composed of the incubation solution with 0.3% v/v DMSO, and the experimental incubation solution has 0.3% v/v DMSO and 50 μM Urolithin A. Urolithin A supplementation was administered using *Ce*Lab, a miniature *C. elegans* Lab-on-chip device designed to carry out a variety of worm assays such as lifespan, reproductive span, and brood size assays at high throughput. This multi-layered microfluidic device is fabricated using standard soft lithography. 200 worms are loaded into separate incubation arenas of each chip. After mating L4 animals on NGM plates for 24 hours, mated Day 1 worms are loaded into *Ce*Lab and are incubated in S-medium with 60 OD600 OP50-1, 50 mg/l streptomycin, and 10 mg/l nystatin. Each animal's reproductive span and lifespan are daily scored using *Ce*Aid, a smartphone application for logging worm assays [65]. After scoring each day, progeny are flushed out of the chip using M9 buffer solution for 20 min and then the incubation solution is replenished.

## Urolithin A treatment oocyte quality assays

L4 worms were mated on NGM plates seeded with OP50 for 24 hours. On Day 1 of adulthood hermaphrodites were separated from males and transferred to NGM plates that were prepared with either 0.3% v/v DMSO or 0.3% v/v DMSO and either 50 μM (or 100 μM) Urolithin A. Hermaphrodites were moved to fresh DMSO control or DMSO plus Urolithin A plates every 2 days until Day 6 of adulthood. Oocyte quality imaging was performed as described above on Day 6 of adulthood.

## Quantification and statistical analysis

Lifespans and reproductive span assays were assessed using the standard Kaplan-Meier log rank survival tests. The first day of adulthood was defined as t = 0. Oocyte quality assays used a chi square test to determine whether there was any significant difference between populations for each oocyte quality category. Embryonic lethality assays used a two-way Anova to compare across experimental days and genotypes. In late-mating assays chi-square tests were used to compare percentages of reproductive and non-reproductive populations across genotypes. Error bars represent 95% confidence interval. For morphology analysis, chi-square tests were used to compare the percentages of the population within each morphology category.

All experiments were repeated on separate days with separate, independent populations. Details of each represented experiment, including sample sizes, can be found in the figure legend. All figures in the Article and Supplement shown are of 1/3 (or in some cases 1/2) biological replicates, with the exception of the mitochondrial morphology scoring, which are pooled data from 3 independent experiments. Prism 9 software was used for statistical analysis.

## Supporting information

**S1 Fig. Mitochondrial morphology in N2 and *daf-2* oocytes and muscle.** (A) Day 1 adult mitochondrial morphology images in the -1 oocytes of N2 (left 2 panels) and *daf-2(e1370)* (right 2 panels). (B) Day 7 adult mitochondrial morphology images in the -1 oocytes of N2

(left 2 panels) and *daf-2(e1370)* (right 2 panels). In A and B images were taken on the Nikon eclipse Ti at 60x magnification. (C) Day 7 adult mitochondrial morphology images in the muscle of N2 (left 2 panels) and *daf-2(e1370)* (right 2 panels). In C images were taken on the scanning confocal Nikon A1 at 60x. All images are of mitochondria stained with ATP5α to mark membranes.
(PDF)

**S2 Fig. Mitochondrial dynamics are specifically required for reproduction function.** (A) Lifespans are not affected by loss of *drp-1(tm1108)* (n = 81, N2 n = 81) (B) *fzo-1(tm1133)* (n = 80, N2 n = 83). (C) Progeny count of mated N2 (n = 12), *drp-1* (n = 13) and *fzo-1* (n = 12) after mating (L4-Day 1) and after mating confirmation (Day 1-Day 2). Day 3, 4 and 5 of adulthood were scored. 2-way ANOVA.
(PDF)

**S3 Fig. Fission, not fusion, is required for *daf-2* extended reproductive longevity.** (A) Representative images of young *daf-2;drp-1* and *daf-2*.Scale bar 20μm. (B) Representative images of aged *daf-2(e1370)* left and *daf-2(e1370);fzo-1(tm1133)* germlines. Scale bar 50μm. (C) Reproductive spans of *daf-2* vs. *daf-2;drp-1* (Fig 3D) including N2 and *drp-1* genotypes. (D) Reproductive spans of *daf-2* vs. *daf-2;fzo-1* (Fig 3E) including N2 and *fzo-1* genotypes.
(PDF)

**S4 Fig. PINK-1 mitophagy is specifically required for *daf-2* reproductive span.** (A) mtRosella is silenced in the germline. Image of the dissected germline of the WMB1119 strain with P*pie-1*TOMM-20::Rosella insert. Image was captured on the Nikon eclipse Ti at 60x magnification. The mCherry and FITC channels are superimposed Z-stacks taken at 0.7μm steps. (B) Mating capability of all genotypes used in this study when young hermaphrodites are mated with *fog-2* males. (C) Late-mating capability in *pink-1* vs. N2. Day 7 adults are mated with young males. (D) Lifespan of *pink-1(tm1779)* (n = 62) is unchanged compared to N2 (n = 69).
(PDF)

**S5 Fig. UA treatment in aged adults.** (A) Lifespan of N2 (unmated) hermaphrodites is extended with Urolithin A treatment (n = 100) vs. 0.3% DMSO (n = 100). Worms were cultured on NGM plates seeded with OP50 during development. Treatment started with or without Urolithin A on Day 1 and loaded in the CeLab chip. (B) *daf-2* reproductive span does not improve with Urolithin A treatment. DMSO control (N = 50) vs. UA treated (n = 50).
(PDF)

**S1 Table. Statistical data tables.** (A) Reproductive span, (B) Late-mating, and (C) Lifespan statistical information. Statistics were performed on Prism. Mean reproductive/alive was performed on OASIS.
(DOCX)

**S1 Video. Representative Z-stack of mitochondria in the -1 oocyte of Day 7 adult N2, 100x.**
(AVI)

**S2 Video. Representative Z-stack of mitochondria in the -1 oocyte of Day 7 adult N2, 60x.**
(AVI)

**S3 Video. Representative Z-stack of mitochondria in the -1 oocyte of Day 7 adult N2, 60x.**
(AVI)

**S4 Video. Representative Z-stack of mitochondria in the -1 oocyte of Day 7 adult *daf-2* (e1370), 100x.**
(AVI)

**S5 Video. Representative Z-stack of mitochondria in the -1 oocyte of Day 7 adult *daf-2* (e1370), 60x.**
(AVI)

**S6 Video. Representative Z-stack of mitochondria in the -1 oocyte of Day 7 adult *daf-2* (e1370), 60x.**
(AVI)

## Acknowledgments

We thank the *Caenorhabditis* Genetics Center (CGC) and the National BioResource Project (NBRP) for strains, Jasmine Ashraf, William Keyes, and Rebecca S. Moore for help with experiments, members of the Murphy Lab for discussion and feedback on the manuscript, Biorender.com for model figure design software. Imaging was performed with support from the Confocal Imaging Facility, a Nikon Center of Excellence, in the Department of Molecular Biology at Princeton University.

## Author Contributions

**Conceptualization:** Vanessa Cota, Coleen T. Murphy.

**Formal analysis:** Vanessa Cota, Salman Sohrabi.

**Funding acquisition:** Coleen T. Murphy.

**Investigation:** Vanessa Cota.

**Methodology:** Vanessa Cota, Salman Sohrabi, Rachel Kaletsky.

**Project administration:** Coleen T. Murphy.

**Supervision:** Coleen T. Murphy.

**Validation:** Vanessa Cota.

**Visualization:** Vanessa Cota.

**Writing – original draft:** Vanessa Cota, Coleen T. Murphy.

**Writing – review & editing:** Vanessa Cota, Coleen T. Murphy.

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
