## [Decision Letter · Decision Letter 0]

1 Jul 2022

Dear Dr Murphy,

Thank you very much for submitting your Research Article entitled 'Oocyte mitophagy is critical for extended reproductive longevity' to PLOS Genetics.

The manuscript was fully evaluated at the editorial level and by independent peer reviewers. The reviewers appreciated the attention to an important problem, but raised some concerns about the current manuscript, especially regarding controls in several key experiments. Based on the reviews, we will not be able to accept this version of the manuscript, but we would be willing to review a revised version. We cannot, of course, promise publication at that time.

If you decide to revise the manuscript for further consideration at PLOS Genetics, please aim to resubmit within the next 60 days, unless it will take extra time to address the concerns of the reviewers, in which case we would appreciate an expected resubmission date by email to plosgenetics@plos.org.

[LINK]

We are sorry that we cannot be more positive about your manuscript at this stage. Please do not hesitate to contact us if you have any concerns or questions.

Yours sincerely,

Arjumand Ghazi

Guest Editor

PLOS Genetics

Gregory P. Copenhaver

Editor-in-Chief

PLOS Genetics

Reviewer's Responses to Questions

**Comments to the Authors:**

Reviewer #1: In the Cota et al manuscript, the authors used the daf-2 mutant in C. elegans as a model to study the role of mitochondrial fragmentation and increased mitophagy in preserving oocyte quality towards the end of the reproductive period. Overall, the paper reported a number of interesting findings, but additional information is required to solidify the conclusions the authors put forth.

Specific comments:

1) It was previously reported that daf-2 mutants have extended reproductive span, but they also have reduced brood size. In this study, the authors performed the experiments using mated daf-2 mutants. Please include quantification of oocyte quality using unmated daf-2 as control (as in Fig 1C).

2) Are fragmented mitochondria (in oocytes) observed in other mutants with extended reproductive span, e.g., TGF beta mutants?

3) Fig 1B showed late mating capability. It’d be important for the authors to demonstrate the non-reproductive worms were mated successfully.

4) In Figure 2, the ovulation rate and total brood size of drp-1 and fzo-1 mutants need to be shown. These are important controls and would help interpret the observed reduced reproductive span phenotype. (It was previously reported that drp-1 depletion reduces brood size and increases lethality.)

5) In Figure 3D, E and FM, it’d be important to include the reproductive span and late mating capability of drp-1 and fzo-1 single mutants as controls.

6) Figure 2 showed that day-6 drp-1 and fzo-1 mutants develop severe oocyte aberration. These phenotypes should also be assessed for the daf-2 and daf-2; fzo-1 mutants in Figure 3 G, H and I.

7) For Figure 3J, the data for unhatched oocytes in daf-2;fzo-1 and single mutants, drp-1 and fzo-1, need to be included as appropriate controls.

8) In Figure 4, reproductive span data for daf-2 and daf-2;pink-1 should be included in parallel to late mating capability data in figure 4A as these two experiments have been shown together in most of the figures to measure reproductive span and oocyte quality.

9) In Figure 4, pink-1 single mutant data for oocyte quality check (uterine mass quantification), reproductive span and late mating capability need to be included as appropriate controls.

(10) The extended reproductive span of daf-2 worms depends on punctate mitochondrial morphology and mitophagy in the oocytes. In figure 4, it is shown that blocking mitophagy by depleting PINK-1 causes severe oocyte anomalies in day-8 daf-2 worms. These results suggest that pink-1 might be overexpressed or mitophagy is increased in the oocytes of daf-2 mutants. It would be important to include pink-1 expression data in daf-2 oocytes.

(11) In figure 5, the effect of Urolithin (a mitophagy promoter metabolite) on reproductive span of daf-2 mutant should be included as a control.

(12) In figure 5, it is shown that UA supplementation reduces the severity of uterine mass in fzo-1 mutants. Interestingly, drp-1 mutants also show severe uterine mass in day-6 worms (figure 2G). The authors should check the effect of UA supplementation in drp-1 mutants to specifically test whether the increased mitophagy upon excessive fragmentation (achieved by fzo-1 depletion) is the underlying mechanism for better oocyte quality in daf-2 mutants. If UA supplementation also reduces the severity of uterine mass in drp-1 mutants, then it would suggest a more general role of UA in regulating oocyte quality irrespective of the mitochondrial morphology in the oocytes.

(13) In Figure 5, the oocyte quality data upon UA supplementation in daf-2, daf-2;drp-1 and daf-2;fzo-1 on day 6 worms should be included as controls.

Minor:

In line 261, Figure 6F-G should be corrected to 5F-G.

Reviewer #2: With the first publication about daf-2 mutants living longer than wild type animals was a finding that was buried in the original paper reporting that these animals also reproduced for a longer period of time. Until the manuscript by Cota et al, the mechanism was unknown. The authors of this manuscript first find that the mitochondria found in the germline of daf-2 mutant animals are more punctate and the extended reproductive profile depends on active mitophagy. This is truly surprising because the mitochondria of the soma are elongated and trend towards fusion, rather than fission.

The authors do an excellent job of teasing apart the role for mitophagy in this process, using SKI-LODGE to assess mitophagy specifically in the germline, showing that pink1 is require for reproductive longevity, but not somatic longevity and finally, ectopic activation of mitophagy can increase the reproductive lifespan of wild type animals.

I am a huge fan of this paper and feel that no new experiments will strengthen this already strong paper. It is suggested that the authors qualify the use of Urolithin A in their studies as not solely targeting the germline since it is drug and hits all tissues.

Reviewer #3: In this very interesting manuscript, Cota et al delved into to the mechanisms of mitochondrial dynamics in regulating female reproductive longevity. Using daf-2 mutated C. elegans as a model, the authors found daf-2 mutant could extend reproductive span through shifting toward fission and activating mitophagy program to maintain oocyte quality. More importantly, they showed reproductive span could be extended by Urolithin A treatment. The present paper is original, well performed, complete, and thoughtfully interpreted. The data provide a novel approach to promote oocyte health. It would strengthen the manuscript if the authors could address the following:

Major points:

To clarify the relationship of daf-2 mutant, fission/fusion, and mitophagy, a few experiments could be conducted.

1. To support daf-2 mutant could drive a shift towards fission and activate/strengthen the mitophagy program, it would be very feasible to compare the drp-1/fzo-1 gene expression ratio and pink-1 gene expression level in oocyte from daf-2 mutant and WT C. elegans.

2. In addition to abrogating the fission/fusion, the approach that the authors used, mimicking punctate/elongated morphology would further support that fission-mediated punctate is required for mitophagy. For example, knockdown of drp-1 or overexpression of fzo-1 in daf-2 mutant C. elegans would elongate mitochondria and thus confer shorter reproductive span?

Minor concerns:

1. Please add the missing scale bar in Figure 1C, Figure 3G, and Figure 3H.

**Have all data underlying the figures and results presented in the manuscript been provided?**

Reviewer #1: Yes

Reviewer #2: Yes

Reviewer #3: Yes

PLOS authors have the option to publish the peer review history of their article (what does this mean?). If published, this will include your full peer review and any attached files.

Reviewer #1: No

Reviewer #2: **Yes: **Andrew Dillin

Reviewer #3: No

---

## [Decision Letter · Decision Letter 1]

29 Aug 2022

Dear Dr Murphy,

We are pleased to inform you that your manuscript entitled "Oocyte mitophagy is critical for extended reproductive longevity" has been editorially accepted for publication in PLOS Genetics. Congratulations!

Yours sincerely,

Arjumand Ghazi

Guest Editor

PLOS Genetics

Gregory P. Copenhaver

Editor-in-Chief

PLOS Genetics

Comments from the reviewers (if applicable):

Reviewer's Responses to Questions

**Comments to the Authors:**

Reviewer #1: The authors have effectively addressed the comments raised previously.

Reviewer #2: Excellent response to reviewer #1. you went far and beyond what was expected or needed to address the overwhelmingly minor comoments of reviewer 1.

Reviewer #3: The authors have made concerted efforts to constructively address my comments.

**Have all data underlying the figures and results presented in the manuscript been provided?**

Reviewer #1: None

Reviewer #2: Yes

Reviewer #3: Yes

PLOS authors have the option to publish the peer review history of their article (what does this mean?). If published, this will include your full peer review and any attached files.

Reviewer #1: No

Reviewer #2: **Yes: **Andrew G Dillin

Reviewer #3: No

**Data Deposition**

http://datadryad.org/submit?journalID=pgenetics&manu=PGENETICS-D-22-00628R1

**Press Queries**

---

## [Editor Report · Acceptance letter]

13 Sep 2022

PGENETICS-D-22-00628R1 

Oocyte mitophagy is critical for extended reproductive longevity 

Dear Dr Murphy, 

We are pleased to inform you that your manuscript entitled "Oocyte mitophagy is critical for extended reproductive longevity" has been formally accepted for publication in PLOS Genetics! Your manuscript is now with our production department and you will be notified of the publication date in due course.

With kind regards,

Agnes Pap

PLOS Genetics

On behalf of:
